# AUTOCHUNK: AUTOMATED ACTIVATION CHUNK FOR MEMORY-EFFICIENT LONG SEQUENCE INFERENCE

**Xuanlei Zhao[1]**[*], **Shenggan Cheng[1], Guangyang Lu[2], Haotian Zhou[2], Bin Jia[2], Yang You[1]**
[1]National University of Singapore   [2]HPC-AI Technology Inc.

## ABSTRACT

Large deep learning models have achieved impressive performance across a range of applications. However, their large memory requirements, including parameter memory and activation memory, have become a significant challenge for their practical serving. While existing methods mainly address parameter memory, the importance of activation memory has been overlooked. Especially for long input sequences, activation memory is expected to experience a significant exponential growth as the length of sequences increases. In this approach, we propose AutoChunk, an automatic and adaptive compiler system that efficiently reduces activation memory for long sequence inference by chunk strategies. The proposed system generates chunk plans by optimizing through multiple stages. In each stage, the chunk search pass explores all possible chunk candidates and the chunk selection pass identifies the optimal one. At runtime, AutoChunk employs code generation to automatically apply chunk strategies. The experiments demonstrate that AutoChunk can reduce over 80% of activation memory while maintaining speed loss within 10%, extend max sequence length by 3.2x to 11.7x, and outperform state-of-the-art methods by a large margin.

## 1 INTRODUCTION

In recent times, significant progress has been made in large deep learning models, with their remarkable capabilities demonstrated across a range of domains, including natural language processing (e.g., GPT-3 (Brown et al., 2020)), computer vision (e.g., ViT (Dosovitskiy et al., 2021)), multimodal applications (e.g., DALL-E (Ramesh et al., 2022)) and protein prediction (e.g., AlphaFold (Jumper et al., 2021)). As the scale of models increases, the substantial demand for memory resources emerges as a major bottleneck for their application.

Model memory can be distinguished into parameter memory and activation memory. Researchers have made efforts to reduce parameter memory cost through techniques like data movement (Ren et al., 2021; Aminabadi et al., 2022) and parallelism (Shoeybi et al., 2020; Rajbhandari et al., 2020). However, relatively little attention has been given to the activation memory in inference, which refers to the

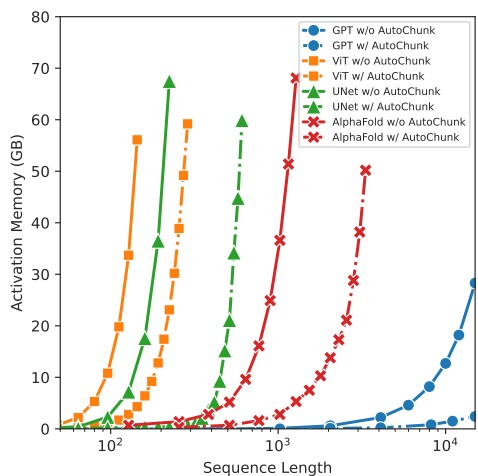

Figure 1: Comparison of activation memory usage with and without AutoChunk.

storage of intermediate tensors during the model's computation. However, with the increasing size and complexity of models, activation memory is becoming a critical consideration for long input sequences, such as documents, images and spatial-temporal information. Activation memory is expected to experience a significant exponential growth as the length of sequences increases, as shown in Figure 1, which makes their inference challenging and costly.

---

[*]This work was done during Xuanlei's internship at HPC-AI Technology Inc.

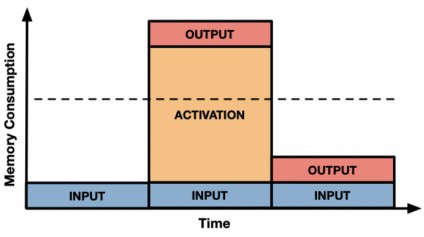 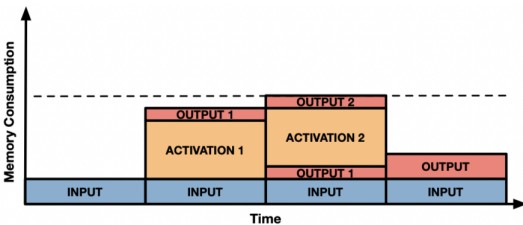

(a) Origin activation memory.    (b) Activation memory with chunk.

Figure 2: Demonstration of activation chunk.

One approach to mitigate activation memory is quantization (Han et al., 2016; Krishnamoorthi, 2018), but its compression capability is limited and and may introduce errors to the results. Another method is fused kernels like FlashAttention (Dao et al., 2022), yet they are primarily tailored for specific modules such as Attention (Vaswani et al., 2017). Recent studies (Jumper et al., 2021; Liu et al., 2022; Kitaev et al., 2020) have proposed chunk as a solution for activation memory in attention and feed-forward. As depicted in Figure 2, chunk decomposes intermediate activation into multiple segments and computes them sequentially to reduce the peak activation memory usage. It represents an effective and promising solution for handling activation memory in more general scenarios. However, there are also several limitations that need to be addressed: 1) Chunk aims to reduce activation memory but may sacrifice computational efficiency, posing speed challenges for real-time serving tasks. 2) Many chunk settings are very speed-sensitive and require careful optimization. 3) Manual chunk design for different models and various input sequences within a model is often suboptimal, falling significantly short of the desired level of efficiency, and is so labor-intensive that it becomes nearly impractical even for experts.

To address the challenges outlined above, we propose AutoChunk, an automatic and adaptive compiler system that efficiently reduces activation memory for long sequence inference. In compiler passes, AutoChunk generates chunk execution plans by optimizing the plan with dynamic programming through multiple stacks of chunk search and chunk selection. It first identifies all potential chunks for the given model based on its analysis on computation graph and memory status, providing flexibility for various chunk settings. Then it employs the chunk selection method to determine the best chunking strategy for the given scenario, striking a balance between speed and memory usage. In runtime, we utilize code generation to apply and recompile the chunk plans to the code automatically. Our experiments shows promising results that AutoChunk reduce 80% of activation memory while maintaining speed loss within 10%, and extend max sequence length by 3.2x to 11.7x. And it significantly surpasses state-of-the-art expert-designed chunk strategy and fused kernels.

We summarize our contributions as follows:

- We introduce an innovative method for enabling automated chunk for long sequence inference to reduce activation memory effectively. Our approach allows for searching and generating chunks of any dimension and settings, making it the first of its kind.

- We propose a novel metric for evaluating the speed loss of different chunks based on our observation of the uneven distribution of memory cost. Through this metric, we can select the best chunk that reduces memory usage significantly while minimizing speed loss.

- We have demonstrated AutoChunk can reduce 80% of activation memory while maintaining speed loss within 10%, and extend max sequence length by 3.2x to 11.7x. AutoChunk surpasses both expert-designed chunk strategies and fused kernels in performance.

## 2 PRELIMINARY AND RELATED WORK

### 2.1 ACTIVATION MEMORY

Activation memory refers to the intermediate tensor memory used during the model's computation in inference. For a module represented as $Y = F(X)$, there are three parts of activation, which

are inputs $X$, outputs $Y$ and intermediate activation $A$. As illustrated in Figure 2a, the cumulative activation memory can be expressed as:

$$M = mem(X) + mem(Y) + mem(A) \qquad (1)$$

However, in contemporary neural networks, the activation memory for long sequence tend to increase rapidly due to three primary factors: 1) The introduction of more complex modules to enhance performance. 2) The adoption of larger models, which results in increased dimension for every tensor. 3) The necessity to process even longer sequences for addressing complex tasks. As depicted in Figure 1, the activation memory demand for models handling long sequences undergoes substantial exponential growth as the sequence length increases, potentially exceeding the parameter memory by several orders of magnitude.

## 2.2 CHUNK

To mitigate the issue of activation memory in attention and feed-forward during inference, the chunk method (Jumper et al., 2021; Liu et al., 2022; Kitaev et al., 2020) has been proposed. To apply chunk to a module represented as $Y = F(X)$, the input $X$ is initially partitioned into $n$ segments denoted as $[x_1, x_2, \ldots, x_n]$, where $n$ is called chunk size. Subsequently, for each segment $x_i$, we calculate its corresponding output using the module, resulting in $y_i = F(x_i)$. Finally, we concatenate all segments' outputs to obtain the final output as $Y = [y_1; y_2; \ldots; y_n]$. As shown in Figure 2b, the activation memory with chunk can be computed as:

$$M = mem(X) + mem(Y) + mem(A)/n, \qquad (2)$$

This process effectively reduces the intermediate activation memory in the computation by a factor of $n$ since the size of intermediate tensors is directly related to the input size. Considering that the intermediate activation constitutes most of the activation memory, the chunk method leads to a significant reduction in activation memory requirements.

However, although chunk is simple and effective, its application is still limited for the following reasons: 1) Chunk inherently reduces activation at the cost of computational efficiency. Inadequately designed chunk can result in significant speed degradation, rendering it unsuitable for most real tasks. 2) Several critical settings of chunk, such as chunk regions, chunk dimensions, and chunk sizes, directly impact its speed performance. Optimal settings for all these parameters are necessary for good speed. 3) The manual crafting of chunk strategies for individual models and the varied input sequences is usually suboptimal, notably failing to achieve the desired efficiency levels. Furthermore, this process is very labor-intensive, often surpassing the capabilities of even human experts. These challenges constrain the practical feasibility of applying chunk methods.

## 2.3 DEEP LEARNING COMPILER

For machine learning compilers such as Tensorflow XLA (Sabne, 2020), TorchInductor and TVM (Chen et al., 2018), optimization techniques like operator fusion and loop tiling have been employed to enhance computational speed. However, these methods tend to overlook activation memory considerations and lack the capability to effectively optimize memory utilization over long-range operators from a global perspective. And Jain et al. (2020) aims to reduce activation memory in training automatically by checkpointing (Chen et al., 2016), but is not applicable to inference.

## 3 AUTOCHUNK

This section outlines the system design of AutoChunk. We begin with the definition of chunk and the system's objectives in Section 3.1. In Section 3.2, we provide an overview of the AutoChunk system. Section 3.3 details our chunk search technique for identifying all potential chunks. In Section 3.4, we present our chunk selection approach for selecting the most effective chunks.

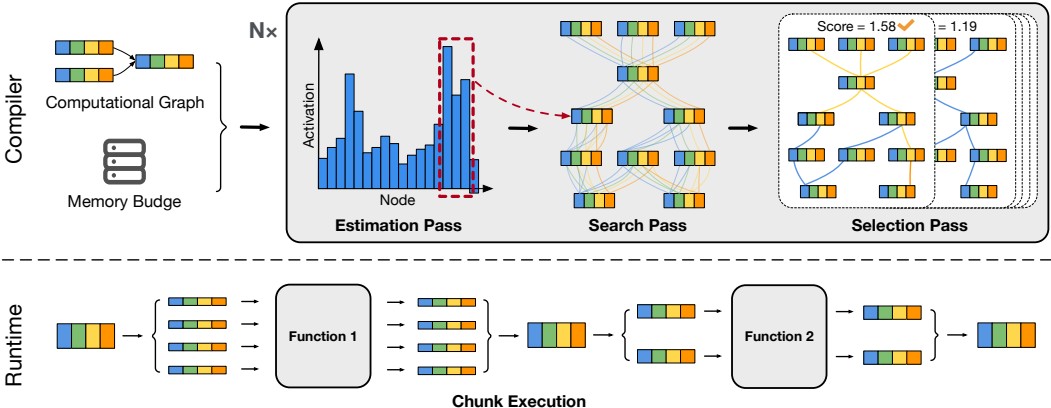

Figure 3: Overview of AutoChunk's compiler passes and runtime architecture.

## 3.1 PROBLEM FORMULATION

We first need to formulate the definition of chunk. Without loss of generalization, considering a neural network $Y = F(X)$, its chunk formulation can be represented as:

$$Y^c = F(X^c, X^{nc}), \tag{3}$$

where inputs $X = [X_1, X_2, ..., X_m]$, outputs $Y = [Y_1, Y_2, ..., Y_k]$. $m$ and $k$ represent the number of inputs and outputs respectively. Chunkable inputs, denoted by $X^c = [X_1^{c_{x_1}}, X_2^{c_{x_2}}, ..., X_n^{c_{x_n}}]$, are inputs that can be divided into chunks for sequential computation; non-chunkable inputs, denoted by $X^{nc} = [X_{n+1}, X_{n+2}, ..., X_m]$, are inputs that cannot be chunked such as leaf nodes and residual; and chunkable outputs, denoted by $Y^c = [Y_1^{c_{y_1}}, Y_2^{c_{y_2}}, ..., Y_k^{c_{y_k}}]$, are outputs that are produced in chunks. Here, $X_n^{c_{x_n}}$ indicates that input $X_n$ is split into chunks along dimension $c_{x_n}$ and computed sequentially, and the number of chunks is called chunk size. In this approach, the goal of AutoChunk is find best chunk regions $F$ and their corresponding settings $X^c, X^{nc}, Y^c$.

## 3.2 OVERVIEW

The aim of this approach is to identify the optimal chunk configuration for any given model, with the goal of controlling activation memory within limit while minimizing speed loss. To this end, we design AutoChunk, which is a compiler system that generates chunk execution plans by optimizing the plan through multiple stacks of chunk search and chunk selection. In chunk search, AutoChunk identifies all possible chunk possibilities in the model, with respect to the given memory budget. In chunk selection, AutoChunk tries to minimize the speed loss for chunk strategy by choosing the optimal chunk from all candidates. Through this optimization process, AutoChunk generates chunk plans for the entire model that meets our requirements for speed and memory.

To achieve this, AutoChunk implements novel compilation passes as Figure 3 illustrates. To be specific, given memory budget and model's computation graph in the form of intermediate representation (IR), AutoChunk generates chunks, leveraging three distinct passes. The estimation pass estimates the activation memory cost and identifies the peak activation memory node for a given computation graph. Then, in the chunk search pass, AutoChunk employs a bottom-up breath-first search algorithm to explore every possible chunk region in the IR, gathering potential chunk strategies. Finally, the chunk selection pass uses dynamic programming (DP) to select the optimal chunk configuration that reduces memory usage and minimizes speed loss. By iteratively repeating the above steps, AutoChunk generates chunks until the memory budget is met.

Given the generated chunk plan, AutoChunk employs code generation based on PyTorch FX (Paszke et al., 2019) and recompile the computation graph with chunk plans. During the runtime pass, the chunk execution is invoked based on the modified computation graph. User can simply call our wrapped function $model = autochunk(model, memory\_budget)$ to apply AutoChunk.

## 3.3 CHUNK SEARCH

In chunk search, AutoChunk utilizes a novel bottom-up breadth-first algorithm to explore the chunk space and identify all possible chunk solutions. Our algorithm can cover strategies of any dimensions and constitute individual solutions into chunk flows for any models, which allows us to efficiently cover the entire chunk space. In the following section, we describe the space of chunk strategies and the design of our algorithm.

***Chunk Flow.*** In order to formulate the definition of legal chunk region, we introduce the concept of chunk flow. Chunk flow refers to the path of chunk dimension where it flows across multiple consecutive nodes in the computational graph, and the dimension the chunk flow passes a node is the node's chunk dimension. A legal chunk flow's inputs and outputs should be able to be divided into several parts and compute sequentially. And importantly, the outputs value should remain the same after chunk, which means that chunk flow may be broken by certain computation or reshape. Following Equation 3, considering functions denoted as $Y = F(X)$ and $Z = G(Y)$, a legal chunk flow can be denoted as:

$$F([X_1^i, ..., X_c^i]) = [Y_1^j, ..., Y_c^j] \equiv Y; \;\; G([Y_1^j, ..., Y_c^j]) = [Z_1^k, ..., Z_c^k] \equiv Z, \tag{4}$$

where $X$, $Y$ and $Z$ are divided and computed into $c$ parts from dimension $i$, $j$ and $k$ respectively, each chunk outputs are equal to original outputs, and a chunk flow passes through dimension $i$ of $X$, dimension $j$ of $Y$ and dimension $k$ of $Z$.

***Chunk Space.*** AutoChunk defines the chunk space by proposing four key rules based on chunk flow. Given a computational graph, there are numerous possible ways to execute the chunk plan. For example, where to chunk, how long should chunk region be, which dimension to chunk for every node, what the chunk size is, and how many chunks there should be. Therefore, AutoChunk will first identify all legal chunks in the chunk space.

To be specific, given a module, we propose the following rules to determine whether it can be a legal chunk: 1) Basic Chunk Rule: The basic requirement is that the function can be transformed into the form our definition for chunk in Equation 3, where part of inputs and all outputs should be chunkable. 2) Output Alignment Rule: The outputs of the function must remain same after chunk. 3) Flow Traceability Rule: For each output, there should be at least one chunk flow which can trace back up to the inputs without any interruption due to reshape or computation, which makes sure that the chunk can be applied to the whole function. 4) Unique Setting Rule: Every node in the function should be passed by chunk flows of same chunk settings. These four rules can be formulated as:

$$Rule\ 1\&2 : F(X^c, X^{nc}) = Y^c \equiv Y, \tag{5}$$

$$Rule\ 3 : \quad \forall Y_i^{c_{y_i}} \in Y^c, \exists X_j^{c_{x_j}} \in X^c, \; s.t.\ ChunkFlow(c_{y_i}, c_{x_j}), \tag{6}$$

$$Rule\ 4 : \quad \forall\, n \in N, \exists!\ ChunkSetting(c_n), \tag{7}$$

where $N$ refers to nodes set of the graph, $ChunkFlow(x, y)$ refers to a chunk flow that passes $x$ and $y$, and $ChunkSetting(x)$ refers to the chunk dimension and size for node $x$.

***Algorithm Design.*** By adhering to the aforementioned criteria, we design the chunk search algorithm, as presented in Algorithm 1. Chunk search algorithm takes computational graph $G$, memory budget $M$ and peak activation node $n_p$ as inputs, and returns all possible chunks. It then searches through node pairs representing the chunk region, which refers to chunk start and end nodes and contains the peak node within the region. Once a chunk region is selected, the algorithm gathers the outputs of the chunk region and iterates through each dimension of the outputs to determine whether it constitutes a legal chunk.

To validate the chunk dimension, the algorithm employs a bottom-up breadth-first search algorithm. Starting from the chunk dimension of the output, the algorithm traces the chunk flow in the directed acyclic computation graph towards the input of the chunk region. When the search algorithm passes a node, it determines the corresponding chunk dimensions for this node based on the chunk flow.

The algorithm then applies the previously mentioned criteria [5;6;7] to check whether this new chunk dimension violates any rules. If this node and its chunk dimension passes the check, it will be added

to the chunk flow, and search continues. If the algorithm successfully reaches the inputs of the chunk region without violating any criteria, it constitutes a legal chunk.

***Complexity Optimization.*** As shown in Algorithm 1, the proposed chunk search algorithm possesses a computational complexity of $\mathcal{O}(N_{node}^3)$. It is evident that number of nodes is paramount to the algorithm's complexity. From our conclusion, there is no need to traverse the entire graph but its neighboring nodes are sufficient. Therefore, we limit the search to a carefully designed local window with a size of $k \ll N_{node}$, which reduces the complexity to $\mathcal{O}(k^2 N_{node})$. Furthermore, It is time-consuming if we opt for a complete graph search for every chunk region. So we propose a two-stage search method. In the first stage, we first examine its inputs and outputs by checking whether any chunk flow exists between them. If true, proceed to the second stage and search all nodes within the chunk region. By this method, our algorithm's complexity now becomes $\mathcal{O}(\zeta k^2 N_{node})$, where $\zeta$ refers to filter passing rate.

***Graph Optimization.*** As stated in Section 3.3, we define legal chunk regions by chunk flow. However, this approach

---

**Algorithm 1:** AutoChunk's chunk search algorithm

> **Input:** Computational graph $G$, memory budget $M$
>              and peak activation node $n_p$
> **Output:** Possible Chunks $C$

1   $C \leftarrow \{\}$
2   **for** $chunk\_region$ in GetNodePairs$(G, n_p)$ **do**
3      // do a bottom up search
4      $output\_nodes \leftarrow$ GetOutNodes($chunk\_region$)
5      // check every chunk dim
6      **for** $chunk\_dim$ in dim($output\_nodes$) **do**
7          $nodes \leftarrow output\_nodes$
8          $cur\_chunk \leftarrow \{\}$
9          **while** $nodes =$ BottomUpBFS($nodes$) **do**
10             $new\_chunk \leftarrow$
                 GetChunk($nodes, cur\_chunk$)
11             **if** $new\_chunk$ satisfy Equ. [5;6;7] **then**
12                $cur\_chunk$.update($new\_chunk$)
13             **else**
14                **break**
15             **end if**
16             **if** reach inputs **then**
17                $C$.update($cur\_chunk$)
18             **end if**
19          **end while**
20      **end for**
21 **end for**

---

may not be optimal under certain circumstances, especially when multiple chunks affect each other. For instance, multiple chunk flows may exist within a chunk region, and only some of them require chunk. However, our algorithm is incapable of distinguishing between them. To address this, when detecting multiple dimension flows within a chunk region, we additionally examine whether some irrelevant flows should be moved out.

## 3.4   CHUNK SELECTION

Chunk selection is aimed to identify the best chunk that meets the memory constraints while minimizing the impact on speed. By using our critical observations as a guide, we employ a primary rule for chunk selection, and define a loss function that accurately measures the speed reduction of any chunk method without having to run actual code and utilize dynamic. We uses dynamic programming (DP) to search for optimal chunk solution globally.

***Memory Estimation.*** AutoChunk prioritizes reducing the activation memory cost of the model during inference. Therefore, for the identified chunks, the first step is to verify if they comply with our memory budget. To achieve this, we combine this chunk's config with all chunks generated before, and use our chunk memory analysis algorithm built on PyTorch to estimate the activation memory usage under such circumstances. To make our estimation accurate, we not only consider the intermediate activation and residual tensors, but also calculate the memory cost due to continuous operation, which may have a large effect on the result.

***Selection Algorithm.*** In light of the potential inter-dependencies among chunks, we propose a dynamic programming (DP) algorithm to search the optimal chunk strategy. Specifically, given a model, we iteratively conduct passes until memory limit is met. Each pass generates a candidate chunk $s_i$. The goal of the algorithms is to find the best chunk strategy $S = [s_1, ..., s_l]$ that can minimize speed loss while satisfying memory cap. To that end, we propose two loss functions from macro and micro perspectives to assess the performance of each chunk.

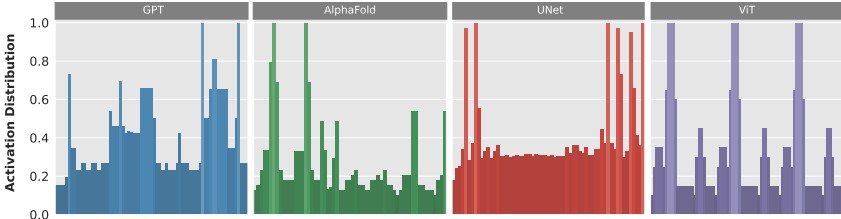

Figure 4: Examples of activation memory distribution. X-axis refers to operators in the model.

Based on our observation, more than 70% of nodes have an activation memory consumption that is less than 30% of the maximum memory usage, as shown in Figure 4. This allows us to achieve an 70% reduction in activation cost by optimizing just 30% of nodes. Therefore, there is no need to chunk an entire module but several consecutive nodes is enough, as the activation memory distribution inside a module is likely to be uneven. To achieve this, we can formulate the macro cost function as:

$$L_{macro} = \alpha N_{node} + \beta N_{flop} \tag{8}$$

where $N_{node}$ is the number of nodes, $N_{flops}$ denotes the total floating-point operations. With this conclusion guide the macro way to chunk, there comes a new question that how we should choose chunk in a micro perspective. The first insight may be we should choose nodes with least ops and node numbers because the less computation is, the less speed will be affected. But our experiment shows that nodes with higher computation density is less likely to be affected by chunk because they still have relatively more parallelism even if the computation is decomposed. And the strides of different chunk dimensions significantly affect the chunk efficiency because of different I/O cost for decomposing and combining tensors, so we should give more weight to dimensions with larger strides. We design the micro cost function as:

$$L_{micro} = \gamma N_{density} + \lambda N_{stride} \tag{9}$$

where $N_{density}$ is computation density calculated as FLOPs per node, and $N_{stride}$ is the stride of the selected dimension. In summary, we need to minimize the cost function as follows:

$$L = L_{macro} + L_{micro} \tag{10}$$

Then we can use this cost function to estimate the performance of every chunk and search the global optimal chunk strategy $S$ with dynamic programming in conjunction with beam search:

$$min \sum_{1}^{l} L(s_i), \quad s.t. \ peak \ memory < memory \ budget. \tag{11}$$

## 4 EVALUATION

This section presents the evaluation of AutoChunk's performance in inference. All experiments are carried out on the NVIDIA Tesla A100 80GB platform with Pytorch. We select GPT (prefill stage), ViT, AlphaFold and UNet (Ronneberger et al., 2015) as our experimental models. UNet refers to the variant used in Stable Diffusion (Rombach et al., 2022), which consists of multiple stacks of ResNet (He et al., 2015) and Transformer (Vaswani et al., 2017) blocks. The hyper parameters of the cost functions in Equations 8 and 9 are automatically tuned.

### 4.1 END-TO-END PERFORMANCE

In this section, we aim to answer three key questions: 1) To what extent does the reduction in activation memory affect speed for AutoChunk? 2) How beneficial is AutoChunk when the fused

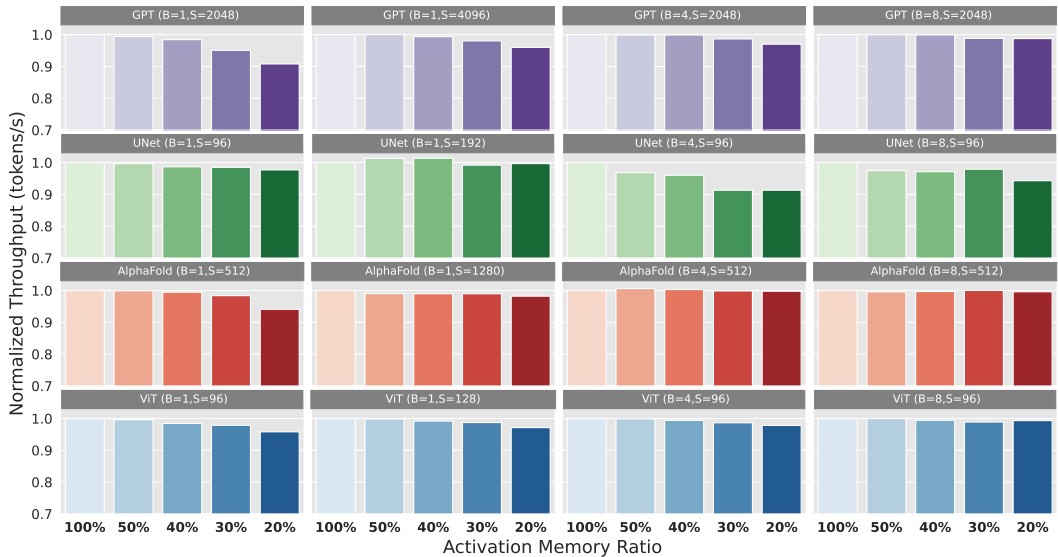

Figure 5: Throughput of AutoChunk for baseline models under various activation memory constraints. The x-axis indicates the ratio of activation memory usage compared to the baseline. The y-axis represents throughput normalized with respect to the baseline.

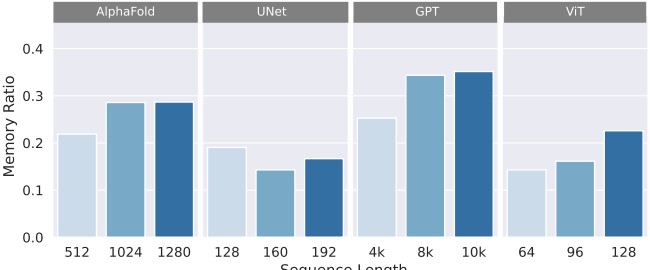

| Strategies | Speed |
|---|---|
| All strategies | 100% |
| No computation density | 84.5% |
| No dimension strides | 75.2% |
| No number of nodes | 89.2% |
| No flops | 91.9% |
| No graph optimization | 67.3% |

Figure 6: The activation memory of AutoChunk and fused kernels, normalized to the memory of fused kernels only.

Table 1: The impact of different strategies on speed.

attention kernel is already used? 3) Does our automated method outperform the expert-designed chunk strategy?

***Performance Against Baseline.*** We evaluate throughput of AutoChunk for baseline models in Figure 5. When utilizing 40% or 50% of the original activation memory, AutoChunk effectively manages to limit throughput loss to within 3%, signifying a negligible impact on speed while effectively halving the activation memory cost for all model types. Furthermore, when operating with only 20% of the original activation memory, AutoChunk's speed reduction experiences a slight increase but remains below 10% which is acceptable for inference. Notably, for larger sequences, we are able to maintain the original speed or even accelerate the inference process. This validates the effectiveness of AutoChunk in reducing activation memory while maintaining speed.

***Performance Against Fused Attention Kernel.*** In contemporary models, the attention mechanism is prevalent, and several fused kernels alleviate the $O(n^2)$ memory cost of it and reduce the peak activation memory. To validate the effectiveness of AutoChunk when fused kernel is already used, we apply memory-efficient attention (Rabe & Staats, 2022) and evaluate AutoChunk's ability to reduce activation memory further. And we control the speed loss of AutoChunk at 5%. As shown in Figure 6, when using fused attention kernels, AutoChunk is able to reduce over 70% of activation memory further at a minor loss in speed. This demonstrates that even when the activation memory of attention is reduced, other parts of the model still maintain high activation memory for long sequence, validating the broad applicability of our approach.

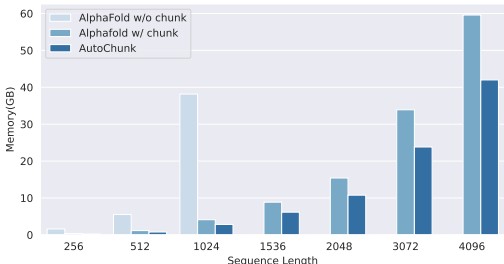 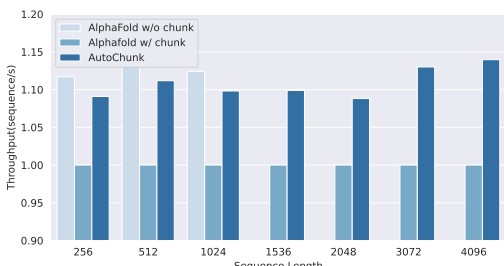

Figure 7: Minimum memory comparison of Expert-Designed chunk and AutoChunk for AlphaFold.

Figure 8: Throughput comparison of expert-designed chunk and AutoChunk for AlphaFold. Throughput is normalized by the former.

***Performance Against Expert-Designed Chunk.*** We compare the throughput of memory cost of origin AlphaFold, AlphaFold with expert-designed chunk from OpenFold (Ahdritz et al., 2022), and AlphaFold with AutoChunk. We compare the minimum memory usage and throughput under same memory limit for various sequence lengths. When the sequence length exceeds 1024, AlphaFold runs out of memory. In Figure 7, AutoChunk can reduce the minimum activation memory usage by 30.6% to 34.4%. In Figure 8, we set the chunk size to 64 for the expert-designed chunk as it's an effective configuration and align the memory cost of them. AutoChunk achieves a speedup ranging from 9.2% to 14.6% when compared to the expert-designed chunk. The results demonstrate that AutoChunk surpasses expert-designed chunk strategies in terms of both speed and memory efficiency.

## 4.2 BREAKING THE MEMORY WALL FOR LONG SEQUENCE INFERENCE

The memory wall has consistently posed a significant challenge for applications involving the processing of long sequences like images and documents. This challenge manifests as a barrier that restricts the execution of models on more economical hardware, characterized by limited GPU DRAM, and on edge devices, which typically possess even scarcer computational resources. As illustrated in Figure 1, our research endeavors have yielded substantial reductions in activation memory usage, achieving reductions spanning several orders of magnitude. Consequently, for 1D inputs of those encountered in models like GPT, our method permits a remarkable 11.7-fold extension in the max inference length. For 2D inputs, such as those encountered in AlphaFold, ViT, and UNet, AutoChunk yields an average 3.2-fold extension in max inference length. This marked improvement significantly mitigates the memory overhead associated with the execution of long sequence inference tasks.

## 4.3 ABLATION STUDY

As illustrated in Table 1, we evaluate the influence of the chunk selection strategy and the graph optimization on system performance. The observed speed metrics represent the average across a range of models and various memory limits. Our findings demonstrate that all considered strategies notably contribute to the system performance, underscoring the sensitivity of speed to these configurations. And graph optimization is effective in reducing redundant computation.

## 5 CONCLUSION

We present AutoChunk, an automatic and adaptive compiler system designed to significantly reduce activation memory usage for long sequence inference through the utilization of chunk strategies. AutoChunk efficiently reduces the activation memory requirements with minor speed loss, while offering a practical solution for deploying models with long sequence on more economical hardware or even edge devices. By effectively reducing memory consumption, AutoChunk enhances the efficiency and accessibility of deep learning applications for long sequence, expanding their applicability in real-world scenarios. In future, AutoChunk can also be adapted to training to reduce activation memory along with checkpointing.

ACKNOWLEDGEMENTS

Yang You's research group is being sponsored by NUS startup grant (Presidential Young Professorship), Singapore MOE Tier-1 grant, ByteDance grant, ARCTIC grant, SMI grant (WBS number: A-8001104-00-00), Alibaba grant, and Google grant for TPU usage.

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
