# OpenReview forum: "AutoChunk: Automated Activation Chunk for Memory-Efficient Deep Learning Inference"
_ICLR.cc/2024/Conference — ICLR 2024 poster_

### Official Review · Reviewer_m8Q2 · 2023-10-28

**Soundness:** 2 fair
**Presentation:** 3 good
**Contribution:** 3 good
**Rating:** 5
**Confidence:** 3

**Summary:**

When serving large-scale deep learning models, their memory requirements are one of the major hurdles. Unlike the parameter memory, optimizations for the activation memory have not been much studied. Since the activation memory is variable depending on the context length, it is important to reduce the activation memory pressure for long context inference. In this research, the authors propose AutoChunk, an automatic compiler system that finds an efficient execution plan with low activation memory pressure. Their evaluation results show that AutoChunk can reduce 50~60% activation memory without speed loss, or 80% activation memory while maintaining speed loss within 10%.

**Strengths:**

- The paper suggests an important problem; optimizing the activation memory because the context length is rapidly increasing.
- Unlike the existence of DL compilers related to parallel execution, the paper presents a new type of DL compiler.

**Weaknesses:**

- Little bit unclear how "activation memory" is measured. Unlike training, we can reuse memory in inference. For example, the MLP module of the Transformer layer has the following structure (not assuming gated linear).
  ```
  Y = UP_PROJ(X)
  Z = DOWN_PROJ(Y)
  ```
  In this case, X and Z can use the same memory region. Did the paper consider such a characteristic? It is confusing because Figure 4 shows the activation memory distribution of each node.
- More analysis for experiments will be helpful. For example, what is a chunking strategy that AutoChunk finds for the GPT model in Figure 5? For now, it is just a black-box compiler.
- For the GPT model, if AutoChunk can reduce the activation memory by half, we can allocate more memory for the key-value cache. It will lead to an end-to-end throughput increase. Are there any results about this? The first paragraph of Section 4 says that the prefill stage is assumed for the GPT case.

**Questions:**

- Could you explain the reason why AutoChunk can even accelerate inference for AlphaFold (B=4, S=512) and UNet (B=1, S=192) cases?
- Is batch dimension also considered as a candidate for chunking? If so, should we run the search algorithm for every execution? It might incur runtime overhead.
- How long does AutoChunk take to search chunking strategy?

---

> ### Author Response · Authors · 2023-12-01
>
> We extend our sincere gratitude for your thoughtful comments and constructive feedback on our paper. We appreciate your time and effort devoted to reviewing our work. In this rebuttal, we aim to clarify and address some of the major points and common concerns raised.
>
> **Q1: How activation is measured?**
>
> **A1**: To assess activation memory in the context of the LLaMA MLP, we can consider the following steps:
>
> ```
> input: x # [b, s, d]
> step1: x_up = self.up_proj(x)  # [b, s, 4d]
> step2: x_gate = self.gate_proj(x) # [b, s, 4d]
> step3: x_gate = x_gate * sigmoid(x_gate) # [b, s, 4d]
> step4: x_down = self.down_proj(x_gate * x_up) # [b, s, d]
> step5: y = x_down # [b, s, d]
> output: y # [b, s, d]
> ```
>
> Assume the shape of the input x is [b, s, d], where b is the batch size, s is the sequence length, and d is the hidden state dimension, and the memory cost of the input x is M.
>
> Then the peak memory cost in LLaMA MLP is in step4, composed of input x, and activation x_up, x_gate, x_gate * x_up, x_down. The total memory cost of them is M + 4M + 4M + 4M + M = 14M. The activation memory here is 13M.
>
> If we use chunk to this module, the peak memory cost in step4 will be input x, output y (because we need to store chunked output), chunked activation x_up, x_gate, x_gate * x_up, x_down. Their total memory cost will be M + M + (4M + 4M + 4M + M) / chunk_size = 2M +  (13M) / chunk_size. Thus, the memory cost can be minimized to 2M, resulting in reduced activation memory.
>
> **Q2: Why autochunk can accelerate inference in some cases?**
>
> **A2**: One reason lies in the cache mechanism in CUDA. After chunking, certain tensors may better fit the cache size, optimizing I/O efficiency and leading to a speedup. And another reason is that the memory cost may reach the GPU capacity, leading to frequent memory allocation and deallocation.
>
> **Q3: How search algorithm works online?**
>
> **A3**: Our search algorithm considers the batch dimension for chunking. We precompute and store search results for various input sizes using a grid strategy, reducing the need for on-the-fly calculations. The grid strategy involves computing input sizes at certain intervals (e.g. every 64 sequence length), considering different combinations of batch size and sequence length. All computed strategies are cached, and selecting the most suitable strategy for a given input from the cache is nearly instantaneous. Thanks to our search algorithm optimizations, the search process takes about 10 minutes with multithreading for most models. And our experiments indicate that the grid cache strategy has minimal impact on speed.
>
> **Q4: Demonstration of chunk strategy?**
>
> **A4**: It is a good suggestion, we will incorporate demonstrations in future versions. Because our chunk solutions differs greatly with different inputs, we here share some key points of our chunk strategy. Take LLaMA model as an example:
>
> 1. For chunk ranges: If the activation is not significant, we primarily chunk softmax in attention and down_proj in MLP. Otherwise, we chunk most of attention, layernorm, and MLP.
> 2. For chunk dimensions: If the batch size allows for chunking (small activation and large batch size), it is the preferred choice. Otherwise, we prioritize chunking based on sequence length, with consideration for hidden state dimension as a last resort.
>
> **Q5: GPT decoding throughput?**
>
> **A5**: During the decoding stage of the GPT model, the activation is relatively small due to the single sequence length per step. Consequently, the impact of our method on decoding throughput is limited.

---

### Official Review · Reviewer_Ypeq · 2023-11-01

**Soundness:** 4 excellent
**Presentation:** 2 fair
**Contribution:** 4 excellent
**Rating:** 8
**Confidence:** 3

**Summary:**

The authors define a formal model for breaking up neural net computations into chunks and sequentially executing them to save on memory footprint. They then formulate it as a search space optimization problem and provide an efficient search algorithm. They show that sequential execution of chunks on a limited set of ops is sufficient to provide good memory use gains while keeping overhead low.

**Strengths:**

- By formalizing the definition of legal chunk flows and providing a cost function, AutoChunk turns a programmer intuition ("certain computations can be sequentialized to save space") into an computational optimization problem. This breaks down of a lot of barriers to entry. Wrapping everything up into a single function call that statically optimizes a compute graph is a testament to just how end-to-end the authors have made their solution.
- Dimension slicing and tiling normally has a stupidly large optimization space. The authors provide several straightforward and effective means for reducing that space to a tractable size and then show that DP is sufficient to get good results.
- The observations on the need for sequential chunking across all operators (fig 4) is useful in understanding the intuition behind why overhead can be kept low. This is generally helpful beyond just its applicability to chunking (even if it has been observed before).
- I appreciated the measure of effectiveness in the presence of other memory optimization (i.e.- fused kernels). Often times, memory optimizations (sparsification, pruning, compression, etc.) partially or fully cannibalize each others benefits when used in conjunction. Good to see these play nicely together.

**Weaknesses:**

- The paper uses a *lot* of bespoke jargon and sometimes uses terms before they are formally introduced. For reference, the following terms are used with the form "chunk ___": flow, region, dimension, size, search, space, setting, selection, formulation, strategy, plan, candidate. If I don't read the word "chunk" again for a while, I'll be happier for it.
- The benefits of AutoChunk vs. expert plans are a bit middling. This is less a weakness of AutoChunk's algorithm and more an observation that the expected benefits from AutoChunk will come from *unchunked* models rather than those already using a chunking strategy. In other words, AutoChunk is more useful when spreading the benefits of chunking to a broader set of models rather than improving on those that already use it.

**Questions:**

- While activation memory is generally correlated with model complexity, chunkable activations seem heavily dependent on the model type. Obviously the models chosen for evaluation in the paper are amenable (which is not a strike against the work---these models are relevant and important). Can the authors give some intuition or generalizations on the classes of neural net architectures that fail to chunk nicely (vs those that do)?

---

> ### Author Response · Authors · 2023-12-01
>
> We extend our heartfelt gratitude for your insightful comments and constructive feedback on our paper. Your acknowledgment of our work is greatly appreciated. We appreciate your time and effort devoted to reviewing our work. In this rebuttal, we aim to clarify and address some of the major points and common concerns raised.
>
> **Q1: Can the authors provide intuition or generalizations on the classes of neural net architectures that fail to chunk nicely (vs those that do)?**
>
> **A1:** Your question is insightful, and I'd like to elaborate from a different perspective: when does autochunk perform poorly?
>
> 1. When activation comprises the majority of memory in a model. If memory dominance is in parameters, optimizing activation offers little assistance in reducing total memory.
> 2. In cases where computation is lightweight. Excessively light computation can result in significant speed loss even with autochunk, as computation parallelism becomes constrained.
> 3. In scenarios where activation cannot be chunked, such as the following examples:
>
> **Example 1: Too many residuals.** Residuals are excessively sparse, and dependencies are long.
>
> ```
> x1 = F1(x)
> x1 = x1 + x
> x2 = F2(x1)
> x2 = x + x1 + x2
> x3 = F3(x1)
> x3 = x + x1 + x2 + x3
> ```
>
> **Example 2: KV-cache in transformer decoding.** These consume a substantial amount of activation memory, but unfortunately, they cannot be chunked.
>
> We hope these examples provide a clearer understanding of situations where autochunk may not perform optimally.
>
> **Q2: The benefits of AutoChunk vs. expert plans are a bit middling.**
>
> **A2:** Thank you for your feedback. In this section, we aim to highlight the efficacy of AutoChunk by comparing its performance with expert-designed chunk strategies in the context of AlphaFold. AlphaFold stands out as the only model meticulously designed with a chunk strategy. In contrast, the other models discussed in our paper lack a well-designed chunk strategy. Our intention is to showcase the generalizability of AutoChunk across these models through baseline experiments. Therefore, for the baseline experiments, we selected models with the most representative modules, including Attention, CNN, MLP, and self-defined modules. We hope this comparison will offer a clearer insight into the advantages of AutoChunk.

---

### Official Review · Reviewer_Tqhc · 2023-11-20

**Soundness:** 3 good
**Presentation:** 3 good
**Contribution:** 4 excellent
**Rating:** 6
**Confidence:** 2

**Summary:**

The paper considers the memory consumption during inference of large deep neural networks on long input sequences. To reduce the activation memory, the paper proposes an adaptive compiler to automatically configure chunking strategies.

**Strengths:**

- The paper tackles an important issue that is becoming increasingly relevant as model sizes continue to grow.

- The empirical evaluation of the proposed method appears thorough.

**Weaknesses:**

- The paper employs substantial jargon and undefined terms. For readers who are not deeply familiar with the topic, sections of the paper are somewhat difficult to comprehend. For instance, it is unclear what portion of activation memory is contained within a chunk.

- The ablation study is arguably somewhat restricted in scope.

**Questions:**

How does the splitting across points inside a batch work?
How exactly does the dynamic programming approach work to solve Equation 11?

---

> ### Author Response · Authors · 2023-12-01
>
> We extend our sincere gratitude for your thoughtful comments and constructive feedback on our paper. We appreciate your time and effort devoted to reviewing our work. In this rebuttal, we aim to clarify and address some of the major points and common concerns raised.
>
> **Q1: How does the splitting across points inside a batch work?**
>
> **A1:** Splitting dimensions directly within a batch may result in computation errors. Therefore, we employ a compiler to identify dimensions that can be split within a batch based on the computation graph. Specifically, we establish policies for each operator and implement the search algorithm outlined in the paper to identify all dimensions eligible for splitting within a batch.
>
> **Q2: How exactly does the dynamic programming approach work to solve Equation 11?**
>
> **A2:** We generate chunk regions one by one, eventually forming our final chunk strategy. However, the order of executing chunks to different parts of nodes may yield varied strategies due to potential overlaps in chunk regions. To address this, we employ dynamic programming to explore all possible orders of chunks, ensuring that we can discover the globally optimal solution.
>
> **Q3: Substantial jargon and undefined terms**
>
> **A3:** Thank you for your feedback; we appreciate your suggestion. In our future revisions, we will strive to enhance the clarity of our work to make it more easily understandable.

---

### Meta-Review · Area_Chair_vyDH · 2023-12-10

**Metareview:**

Model serving is an increasingly important problem; efficient model serving strategies are timely and of interest to the community, and probably dessirves more attention. The reviewers appreciated the experimental evaluation and material presented, but one reviewer found the ablation study limited. Overall, the reviewers also found that the paper could be made more accessible. I would encourage the authors to revise the paper accordingly.

**Justification For Why Not Higher Score:**

Reviewers complained that the paper was difficult to assess as it was using a lot of jargon and could be made more accessible.

**Justification For Why Not Lower Score:**

Reviewers agreed that this was a valuable paper. Efficient model serving is a relevant practical problem and best practices are of interest to the community.

---

### Decision · Program_Chairs · 2024-01-16

Accept (poster)